# Adaptive Construction of Freeform Surface to Integrable Ray Mapping Using Implicit Fixed-Point Iteration

**Jiahua Chen** [1], **Yangui Zhou** [1,2,*], **Hexiang He** [1,2] and **Yongyao Li** [1,2]

[1] School of Physics and Optoelectronic Engineering, Foshan University, Foshan 528000, China; 2112205005@stu.fosu.edu.cn (J.C.); hehexiang@fosu.edu.cn (H.H.); liyongyao@fosu.edu.cn (Y.L.)
[2] Guangdong-HongKong-Macao Joint Laboratory for Intelligent Micro-Nano Optoelectronic Technology, School of Physics and Optoelectronic Engineering, Foshan University, Foshan 528225, China
[*] Correspondence: zygsai@163.com

**Abstract:** Constructing a freeform surface that accurately satisfies both integrable condition and Snell's law under a given invariant source–target map is challenging for freeform design. Here, we propose a fixed-point iteration method to address this problem. This process involves solving a set of balanced gradient equations in the form of fixed-point iterations that are derived from equivalent integrability conditions and Snell's law. By using the convergence theorem of fixed-point iteration, a unique solution for the balanced gradient equations exists, which is determined by the natural geometric properties of the freeform surface and is independent of the mapping. The gradient operators on the left-hand side of the equations are converted into a differential matrix form via a finite difference scheme. In one iteration, differential operations are forward-performed on the right-hand side of the equations, and the system of linear equations is solved on the left-hand side of the equation. The constructed freeform surfaces work well in both the paraxial and nonparaxial regions, and convergence in the nonparaxial region is faster than that in the paraxial region. The robustness and high efficiency of the proposed method are demonstrated with several design examples.

**Keywords:** freeform optical design; balanced gradient equations; implicit fixed point iteration

## 1. Introduction

Freeform optics are increasingly being applied in both imaging and nonimaging optical fields. In nonimaging optics, freeform surfaces are used to collect light energy or reshape light beams [1]. The design methods of freeform surfaces can be classified into three main categories [2]: the Monge–Ampère equation method [2–8], the supporting quadric method [9,10], and the ray mapping method [11–25].

The ray mapping method might be the most popular method for designing freeform surfaces because of its flexibility and simplicity. These methods are generally implemented in two steps. First, the coordinate mapping from the source to the target surface is established, which is also called the source-to-target map. The source-to-target map can be obtained from the energy conservation via the optimal transport scheme with a quadric cost function [12–19]. Secondly, the freeform surface is subsequently constructed according to the mapping.

For freeform surface construction, several methods have been provoked. A conventional method proposed by Wang et al. [11] first builds a seed curve and then constructs the other curves in the orthogonal direction. Due to the independence of constructing direction,

error accumulation occurs in the normal field. To overcome error accumulation, a V1V2 method [19] was proposed to construct a freeform surface by combining two orthogonal tangent vectors perpendicular to the normal vector. However, these pointwise construction methods cannot guarantee exact ray tracing to the target map position in nonparaxial situations because the selection of initial points is arbitrary. To obtain a global solution of a freeform surface, Feng et al. [16] used the chord between two adjacent points perpendicular to the average of the two normal vectors at these two points and enforced this relationship with a least-squares method. However, the approximation of the outgoing vector equal to the target position vector still causes errors in the nonparaxial region. Bösel et al. [17] constructed the freeform surface as a linear advection equation derived from substituting Snell's law into the integrable condition. Similarly, this advection equation is derived from paraxial approximation.

The above surface construction methods, due to the introduction of paraxial or far-field approximations, result in freeform surfaces not meeting integrability conditions [20] in the nonparaxial region:

$$\vec{\mathbf{N}} \cdot (\nabla \times \vec{\mathbf{N}}) = 0, \tag{1}$$

where $\vec{\mathbf{N}}$ denotes the surface normal vector field.

To improve the nonintegrability of the normal vector field in the nonparaxial region, several schemes were proposed. Fournier et al. [20] computed an integrable ray mapping via the supporting quadric method with a data-fitting process. Karel et al. [21] proposed a symplectic flow method to provide intermediate mapping for complex off-axis and nonparaxial illumination problems. Doskolovich et al. proposed a discrete method to transform the illumination design problem into a linear assignment problem [22–24]. Wei et al. [25] proposed a least-squares method to dynamically compute the ray mapping that satisfies the integrable condition in the surface construction process.

The above improved methods [20–25] "push" the integrable conditions into the construction process of mapping to overcome the weakness of using paraxial or far-field approximation in freeform surface construction. In other words, these methods focus on changing the mapping to adapt to surfaces with errors rather than eliminating the approximation in surface construction. Although integrable ray mapping techniques have been well addressed, solving a freeform surface without approximation adaptive to a given invariant mapping remains a challenging problem.

In this paper, we propose a novel freeform surface construction method for collimated beam shaping that works well in nonparaxial situations via implicit fixed-point iteration. Different from other integrable ray mapping methods, the proposed method dynamically iterates the freeform surface to adapt to the mapping without the necessity of changing the mapping during the iteration process. In this method, the differential geometry definition of the normal vector is used as a condition that is equivalent to the integrable condition and is substituted into Snell's law to obtain balanced gradient equations that can be transformed into an implicit expression of fixed-point iteration (FPI). The existence of the solution for this system is demonstrated by the convergence theorem of the FPI method. The mathematical system can be effectively solved via FPI. Interestingly, the farther the deviation of points from the optical axis, the higher their convergence speed. The simulation results show that the constructed freeform surface works well in both the paraxial and nonparaxial regions. We present our freeform optics constructing method detail in Section 2. Several design examples are presented in Section 3. In Section 4, a brief conclusion is presented.

## 2. Freeform Construction Method via Implicit Fixed-Point Iteration

### 2.1. Formulation of Source-to-Target Map

Assume that the source plane is defined in Cartesian coordinates $(x, y)$ at $z = z_0$, and let $E_s$ be the irradiance in a bounded region S. The target plane is defined in coordinates $(t_x, t_y)$ at $z = t_z$, and $E_t$ is the prescribed irradiance distribution in region T. The inverse problem is to design a freeform surface $Z(x, y)$ to redirect the rays $\vec{I}(x, y)$ to the target plane so that we can obtain a predefined irradiance $E_t(t_x, t_y)$. The sketch is presented in Figure 1. As shown in Figure 1, $\vec{O}$ and $\vec{I}$ refer to the unit outgoing and incident light fields, respectively. $\theta$ is the off-axis angle between the outgoing vector $\vec{O}$ and the z-axis.

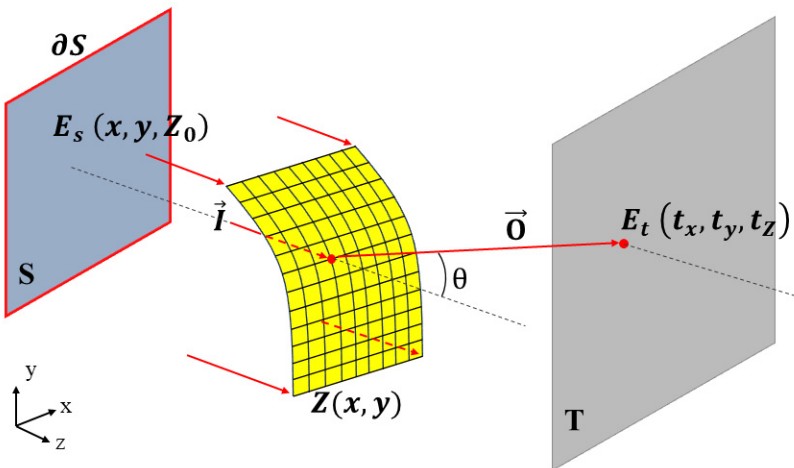

**Figure 1.** The mapping principle of freeform surface and target surface.

According to energy conservation and optimal transport, a map $\Phi : S \rightarrow T$ redistributing the energy density $E_s$ to $E_t$ can be expressed as follows [19]:

$$E_t(\Phi(x, y))\det(\nabla\Phi) = E_s(x, y), \tag{2}$$

where $\det(\nabla\Phi)$ is the determinant of the Jacobian matrix of function $\Phi$.

The transport boundary condition is forced to map the boundary $\partial\Omega_s$ of S to the boundary $\partial\Omega_t$ of T, which can be written as

$$F(x, y, \Phi) = 0 \quad (x, y) \in \partial S. \tag{3}$$

By solving Equations (2) and (3), a source-to-target map $\Phi$ can be obtained. For some efficient numerical solver for Equations (2) and (3), one can refer to [26,27]. After obtaining the map, the critical problem is to find a freeform surface that can accurately transform the light rays from the source to the target region. Below, an effective method for constructing an integrable freeform surface in nonparaxial situations is presented.

### 2.2. Balanced Gradient Equations for Freeform Surfaces

Assume that G is a point on the freeform surface and that its position vector is $\vec{G} = (x, y, Z(x, y))$. The unit normal vector $\vec{N}$ of the freeform surface at point $\vec{G}$ for a collimated beam can be written as

$$\vec{N} = \frac{\partial_x\vec{G} \times \partial_y\vec{G}}{\left|\partial_x\vec{G} \times \partial_y\vec{G}\right|} = \frac{(-\partial_x Z, -\partial_y Z, 1)}{\sqrt{(\partial_x Z)^2 + (\partial_y Z)^2 + 1}}, \tag{4}$$

where $\partial_x \vec{\mathbf{G}}, \partial_y \vec{\mathbf{G}}$ and $\partial_x Z, \partial_y Z$ are the first-order partial derivatives of $\vec{\mathbf{G}}$ and $Z$ with respect to x and y, respectively.

It is easy to prove that Equations (1) and (4) are equivalent in determining whether a surface is smooth or continuous. Note that although Equation (4) is equivalent to Equation (1), the advantage of using Equation (4) is that it does not require calculating the curl of the normal vector during the construction process.

At any point on the surface, Snell's law should be obeyed:

$$A\vec{\mathbf{N}} = n_o \vec{\mathbf{O}} - n_i \vec{\mathbf{I}}, \tag{5}$$

where $A = [n_i^2 + n_o^2 - 2n_i n_o (\vec{\mathbf{O}} \cdot \vec{\mathbf{I}})]^{1/2}, \vec{\mathbf{N}}$ refers to the unit normal vector field, and for refractive indices $n_i$ of the lens and $n_o$ of the surrounding medium. Here, the freeform lens is surrounded in air, so $n_o = 1$.

For collimated beams, by substituting Equation (4) into Equation (5), two balanced gradient equations (BGE) can be obtained:

$$\begin{cases} \partial_x Z = O_x' + O_z' \partial_x Z \\ \partial_y Z = O_y' + O_z' \partial_y Z \end{cases}, \tag{6}$$

where $(O_x', O_y', O_z') = (O_x, O_y, O_z)/n_i$, and $(O_x, O_y, O_z)$ refer to the three components of the unit outgoing light vector $\vec{\mathbf{O}}$ in the x, y, and z directions. For a detailed derivation of Equation (6), see Appendix A.

Snell's law in Equation (5) reveals the relationship between normal vector $\vec{\mathbf{N}}$, outgoing light vector $\vec{\mathbf{O}}$, and incident light vector $\vec{\mathbf{I}}$ at every point on a surface. Equation (6) is derived from Snell's law, so it expresses the relationship between the corresponding physical quantities of surface coordinates Z and outgoing vector $\vec{\mathbf{O}}$ at each point on the surface. The left-hand side of Equation (6) represents the gradient of freeform surface in the incident field, while the right-hand side represents the gradient of freeform surface in the outgoing field, and they are equal to each other. That is why we called Equation (6) as balanced gradient equations for freeform surface.

Since the BGEs were derived from Equation (4), which is an equivalent integrable normal field, the solution of the BGEs should naturally satisfy the integrable condition of Equation (1).

Note that if the paraxial approximation is applied in the right-hand side of BGEs, Equation (6) will degenerate into a similar linear advection equation as in [17]. However, the BGEs in this paper do not omit the right-hand side of Equation (6), which is strongly connected to the nonparaxial outgoing field. Therefore, the freeform surface constructed by BGEs can accurately transmit rays from the source onto the target position in the nonparaxial region.

There are two challenging points to solve the BGEs of Equation (6). First, Equation (6) includes two first-order PDEs. Numerically solving each PDE in Equation (6) may yield two different groups of data points. Secondly, the outgoing field $\vec{\mathbf{O}}$ on the right-hand side of Equation (6) is nonlinear and dependent on Z; a suitable algorithm is needed to handle this problem. We propose an implicit fixed-point iteration (IFPI) process to deal with these problems.

*2.3. Existence of Solution for BGE via FPI Convergence Theorem*

Below, we explore whether there is a feasible solution for the BGEs in Equation (6) for a given source-to-target map $\Phi^*$.

The distance between the freeform surface and the target plane along the outgoing vector $\overrightarrow{\mathbf{O}}$ is labeled as $\rho$. The distance functional $\rho(x, y, z)$ can be expressed as follows:

$$\rho = \sqrt{(t_x - x)^2 + (t_y - y)^2 + (t_z - Z)^2}. \tag{7}$$

Combining Equation (6) with the differential geometry relationship, the relationship between the freeform surface coordinates Z and the distance functional $\rho$ can be obtained as follows (for detailed derivations, see Appendix B):

$$Z = C - \rho/n_i \tag{8}$$

where C is a constant.

Because Z and $\rho$ are both unknown variables in Equation (7), it is difficult to solve them using just a single equation. However, Equation (8) has the form $Z = \varphi(Z)$, which is usually the standard form of FPI. When a unique convergent solution for $Z = \varphi(Z)$ can be obtained by using FPI, the following theorem must be satisfied [28]:

**Theorem 1.** *Let the iterative function $\varphi(Z)$ be continuous in the range of $[a, b]$, and let it satisfy the following:*

*(I)    When $Z \in [a, b]$, then $a \le \varphi(Z) \le b$;*

*(II)    There is a positive number $L, 0 < L < 1$, and $\forall Z \in [a, b]$, which satisfies $|\partial_z \varphi| < L$.*

Then, $Z = \varphi(Z)$ has a unique solution $Z^*$ in $[a, b]$, and for any initial value $Z_0$, iteration $Z_{k+1} = \varphi(Z_k)$ converges to $Z^*$.

For Equation (8), let $\varphi(Z) = C - \rho/n_i$, the lower bound $a = 0$, and the upper bound $b = t_z$. Substituting Equation (8) into the two conditions of Theorem 1, we can obtain the following:

*(I)    When $Z \in [0, t_z], \rho \le n_i C \le \rho + n_i t_z$;*

*(II)    $|\partial_Z \varphi| = O_Z / n_i < 1$.*

Equation (8) clearly satisfies the converging theorem of FPI by properly choosing the constant $C$. For condition (I), $n_i C$ should be chosen in a range of $\rho \sim \rho + n_i t_z$. As long as $t_z > 0$, there must be a suitable C that satisfies this condition. This can be easily achieved by setting the target surface at a position greater than zero.

For condition (II), $\partial\varphi/\partial Z = O_z/n_i$. $O_z$ is the z component of the unit outgoing vector $\overrightarrow{\mathbf{O}}$ and always satisfies $O_z \le 1$; $n_i$ is the refractive index of the freeform lens, and $n_i > 1$. Therefore, $O_z/n_i < 1$ should be satisfied everywhere on the freeform surface. Furthermore, if $O_z$ is smaller, the converging speed of FPI is higher, which means that, for points that deviate farther from the optical axis (with a larger off-axis angle $\theta$), the convergence speed should be higher than that with a smaller off-axis angle (the paraxial region).

It was shown above that Equation (8) has a unique numerical solution $Z^*$ for any initial value $Z_0$ via FPI. Since Equation (8) is the integral form of BGEs of Equation (6), the unique $Z^*$ is the solution of BGEs as well.

### 2.4. IFPI Process for BGE

Because the selection of parameter C has certain arbitrariness, it is not convenient to directly use FPI for Equation (8). Equation (8) shares the same solution of the BGE in Equation (6). Therefore, we find the solution of Equation (6) via the IFPI process.

The explicit expression of BGEs in Equation (6) calculates the differential of Z, while Z itself is implicit. Instead of calculating the integrals of $\partial_x$ and $\partial_y$ in Equation (6), finite difference schemes are applied to the differential operator on the left-hand side of Equation (6)

to obtain Z, whereas differential operators $\partial_x$ and $\partial_y$ are forward-operated to obtain $\partial_x Z$ and $\partial_y Z$ on the right-hand side of Equation (6).

For a rectangular grid with m rows and n columns, the values of Z are located in an $m \times n$ matrix $\mathbf{Z}_M$. Matrix $\mathbf{Z}_M$ is cast as vector $\mathbf{Z}_v$ such that the $i + n(j − 1)$ element of $\mathbf{Z}_v$ is equal to $(\mathbf{Z}_M)_{ij}$ ($\mathbf{Z}_M$ is enumerated column-wise) [29]. We apply the second-order finite difference scheme to the differential operator $\partial_x$ and $\partial_y$ on the left-hand side in Equation (6), then two differentiation matrices $\mathbf{H}_x$ and $\mathbf{H}_y$ with the size of $mn \times mn$ can be obtained. For details of differentiation matrices constructing routine, the reader can refer to [30,31]. Because the two BGEs share a common solution of $\mathbf{Z}_v$, we combine the two differentiation matrices $\mathbf{H}_x$ and $\mathbf{H}_y$ into one matrix as $\mathbf{H} = [\mathbf{H}_x; \mathbf{H}_y]$, which is a $2\,mn \times mn$ matrix with $\mathbf{H}_x$ above and $\mathbf{H}_y$ below.

For the right-hand side of Equation (6), a column vector $\mathbf{B}$ with $2\,mn$ elements can be constructed as follows:

$$B(i) = \begin{cases} O_x' + O_z' \partial_x Z & i \leq mn \\ O_y' + O_z' \partial_y Z & i > mn \end{cases}. \tag{9}$$

After the above discretization processing, the BGEs of Equation (6) are transformed into a linear equation system as follows:

$$\mathbf{H}\mathbf{Z}_v = \mathbf{B}. \tag{10}$$

Equation (10) is a system containing $2\,mn$ linear equations with $mn$ unknown elements of $\mathbf{Z}_v$. By combining the two coefficient matrices of BGEs into a single matrix $\mathbf{H}$, Equation (10) utilizes a common set of $\mathbf{Z}_v$, thereby resolving the issue of having two different data points as presented in Equation (6). Since the coefficient elements in $\mathbf{H}$ are derived from approximate finite difference schemes of differential operators, they remain unchanged during the iteration process. Here, a standard central difference scheme of second order is used for inner points, and a second-order upwind differential scheme is used for boundary points.

The second problem in solving BGEs is that $\mathbf{O}'$ in $\mathbf{B}$ is nonlinear and unknown; here, we treat vector $\mathbf{B}$ as a variable parameter in the IFPI process and rewrite Equation (10) into a more straightforward IFPI form:

$$\mathbf{H}(\mathbf{Z}_v)_{k+1} = (\mathbf{B})_k, \tag{11}$$

where $(*)_k$ represents the kth iteration value of $*$.

Figure 2 illustrates the construction process of freeform surfaces in IFPI. The gradients $\nabla \mathbf{Z}$ on the left-hand side of Equation (6) are attached on the freeform surface in the incident light field, as shown in the below region and labeled as blue arrows in Figure 2. The right-hand side of Equation (6), equaling to Equation (9), is labeled as $\mathbf{B}$ with red arrows attaching on the freeform surface in the outgoing light field.

In the k-th iteration, $(\mathbf{O}')_k$ and $\nabla(\mathbf{Z})_k$ are calculated and substituted into Equation (9) to form the k-th vector $(\mathbf{B})_k$; then, by solving Equation (11), a new freeform surface $(\mathbf{Z}_v)_{k+1} = \mathbf{H}^{-1}[(\mathbf{B})_k]$ can be obtained. In the next iteration, the $(k + 1)$-th value $(\mathbf{Z}_v)_{k+1}$ is substituted into Equation (9) to obtain a new $(\mathbf{B})_{k+1}$ value; by solving Equation (11) again, surface $(\mathbf{Z}_v)_{k+2}$ can be acquired. If the freeform surface reaches the unique solution, the gradients the in incident light field should be balanced to the ones in the outgoing light field with $\nabla(\mathbf{Z})_M = (\mathbf{B})_M$. For an actual criterion in IFPI, we can use $\| (\mathbf{Z}_v)_{M+1} − (\mathbf{Z}_v)_M \| < \delta$, where $\delta$ is the solution accuracy; then, $(\mathbf{Z}_v)_M$ should be the solution $Z^*$. The detailed steps of the IFPI algorithm are shown in Figure 3.

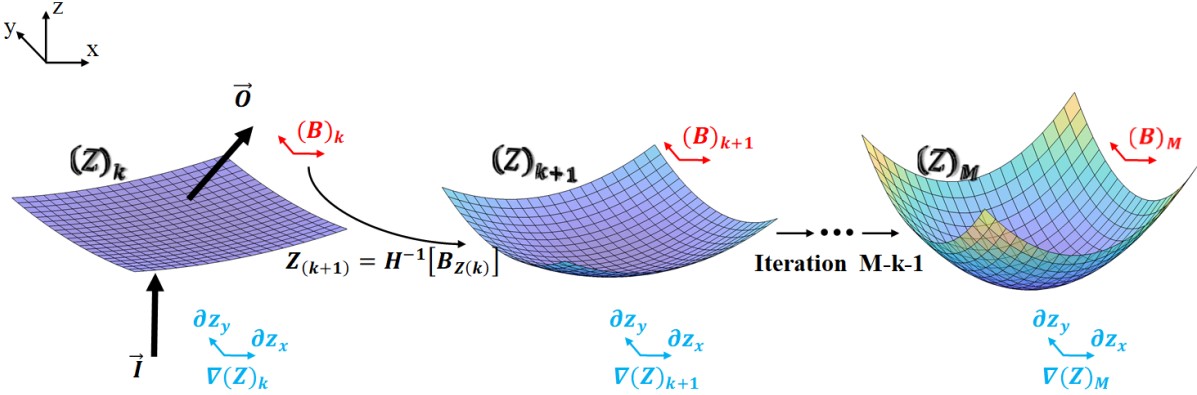

**Figure 2.** Sketch of IFPI process of Equation (11).

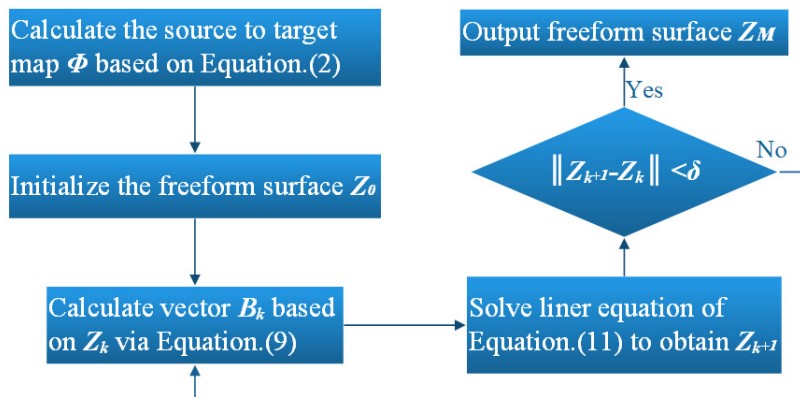

**Figure 3.** The flow chart of IFPI algorithm.

## 3. Design Examples

### 3.1. Convergence Characteristics in IFPI Freeform Construction

The first example explores the feasibility and convergence of the FPI method in freeform optical design. In this example, the meshes (a $50 \times 50$ grid) of the light source and the target surface are set to a uniform distribution with the same unit square area of $1 \text{ mm} \times 1 \text{ mm}$, and the distance between the light source and the target plane is maintained as 10 mm. The position of the light source grid is fixed at the origin, the center position of the target surface is moved from the origin along the Y-axis to increase the off-axis angle $\theta$, and the corresponding freeform surfaces with an increasing off-axis angle are calculated via iterations (8) and (10). The final solution tolerance is set to $\delta = 10^{-6}$, the convergence steps of iterations (8) and (10) are marked for different off-axis angles $\theta$, and the trend of change is shown in Figure 4.

Figure 4a illustrates the variation in the tolerance to convergence steps for IFPI with different off-axis angles. The maximum convergence time of five different $\theta$ is 8.9 s. Figure 4a shows that a greater off-axis angle results in a larger initial tolerance, implying a more challenging precision requirement for surface construction. However, as the number of iterations increases, the tolerances of various angles decrease, and all the tolerances approach $10^{-2}$ after 12 steps. For the iterations for larger off-axis angles, the slopes of the convergence curves are steeper, indicating faster convergence. For example, at the 2nd step, the convergence rate for the off-axis angle $\theta = 42.59°$ is significantly greater than that for $\theta = 19.47°$, approaching the solution at a higher speed. Figure 4b presents the relationship between the off-axis angles and total convergence steps for the same final tolerance of $10^{-6}$. The maximum convergence time of 12 samples is 21.7 s. The diagram indicates a downward trend in the convergence steps as the off-axis angle increases. Specifically, when

the off-axis angle is small, such as 10°, the number of convergence steps is relatively large, implying that more iterations are necessary to achieve design accuracy. As the off-axis angle increases, the number of convergence steps decreases, with a notable reduction at 40°.

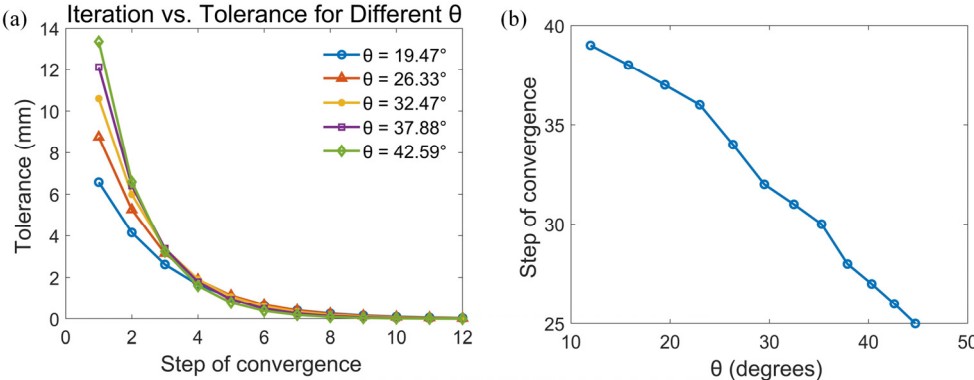

**Figure 4.** (**a**) Relationship between the step of convergence and tolerance with different off-axis angles; (**b**) relationship between the total number of converging steps and off-axis angle with a final tolerance of $10^{-6}$.

Figure 4 confirms that for points that deviate farther from the optical axis, the convergence speed is higher than for those with a smaller off-axis angle (the paraxial region). This is determined by the convergence characteristics of FPI. Because the off-axis angle θ is larger, $O_z = \cos\theta$ is smaller, resulting in faster convergence. This means that the algorithm is suitable for freeform surface construction, especially for nonparaxial regions.

The root-mean-square (RMS) value is used to evaluate the deviation between the precise ray tracing map $\Phi_s$ of the constructed surface and the preset target map $\Phi^*$:

$$\text{RMS} = \sqrt{\frac{1}{mn}\left(\sum_i \sum_j |\Phi_s(i,j) - \Phi^*(i,j)|^2\right)}, \tag{12}$$

where m and n denote the grid number in the $t_x$ and $t_y$ direction.

To test the robustness of the algorithm, we introduced 50 sets of random jitters to the target grid. One set of random grids is shown in Figure 5a. As the number of iterations increases, the RMS quickly decreases. Additionally, different random grids show almost the same convergence trend, as illustrated in Figure 5b. After 50 sets of random grids reached the RMS threshold, the final RMS values were distributed within a stable range, as illustrated in Figure 5c.

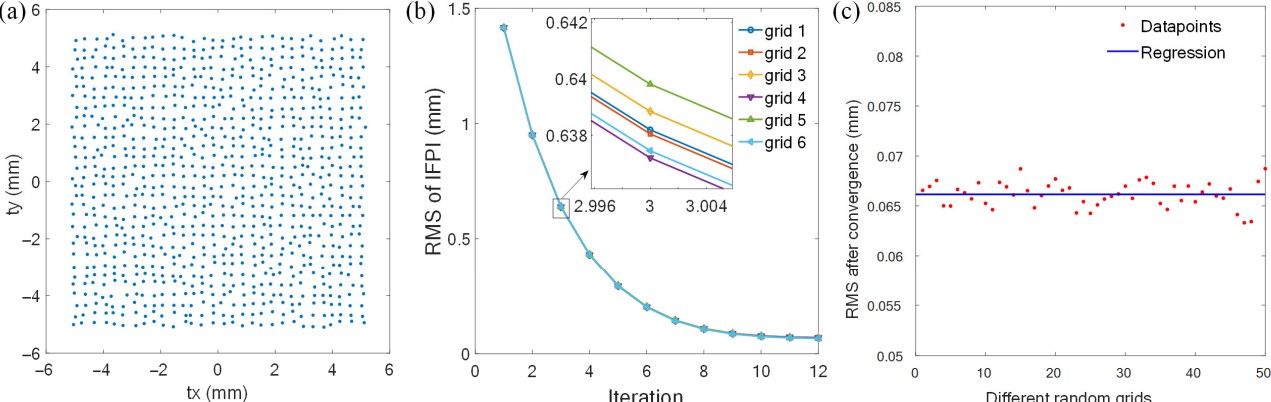

**Figure 5.** Robustness testing. (**a**) One set of 50 random grids; (**b**) RMS decreases with the number of iteration steps for 6 sets of random grids; (**c**) the final RMS distribution of 50 random grids.

### 3.2. Comparison with Other Freeform Surface Construction Methods

To evaluate the efficiency of the IFPI method in constructing freeform surfaces under an unchanged ray mapping, the constructing methods of freeform surface mentioned in the Introduction are recalled for comparative analysis. The aperture of lens in this example is a square with a size of 1 mm × 1 mm, the target distribution is set to be uniform with a size of 50 mm × 50 mm, and the distance between the freeform surface and the target plane is 25 mm. The maximum off-axis angle is about 45° at the four corners of the square.

Figure 6a–d illustrate the accurate ray mapping on the target plane, corresponding to the results of the freeform surface obtained through the V1–V2 [19], linear advection equation [17], conventional surface construction [11], and Feng's methods [16], respectively. Figure 6i corresponds to the result of the freeform surface obtained via the IFPI method. The precise mapping is calculated through our self-written ray mapping program. A comparison of Figure 6a–d,i demonstrates that the surface constructed using IFPI can accurately direct light to the predetermined target grid. In contrast, the surfaces created by the four other methods emit light that deviates from the intended target position, particularly in the nonparaxial region. This deviation becomes more pronounced as the off-axis angle increases, as illustrated by the four edges in Figure 6a–d.

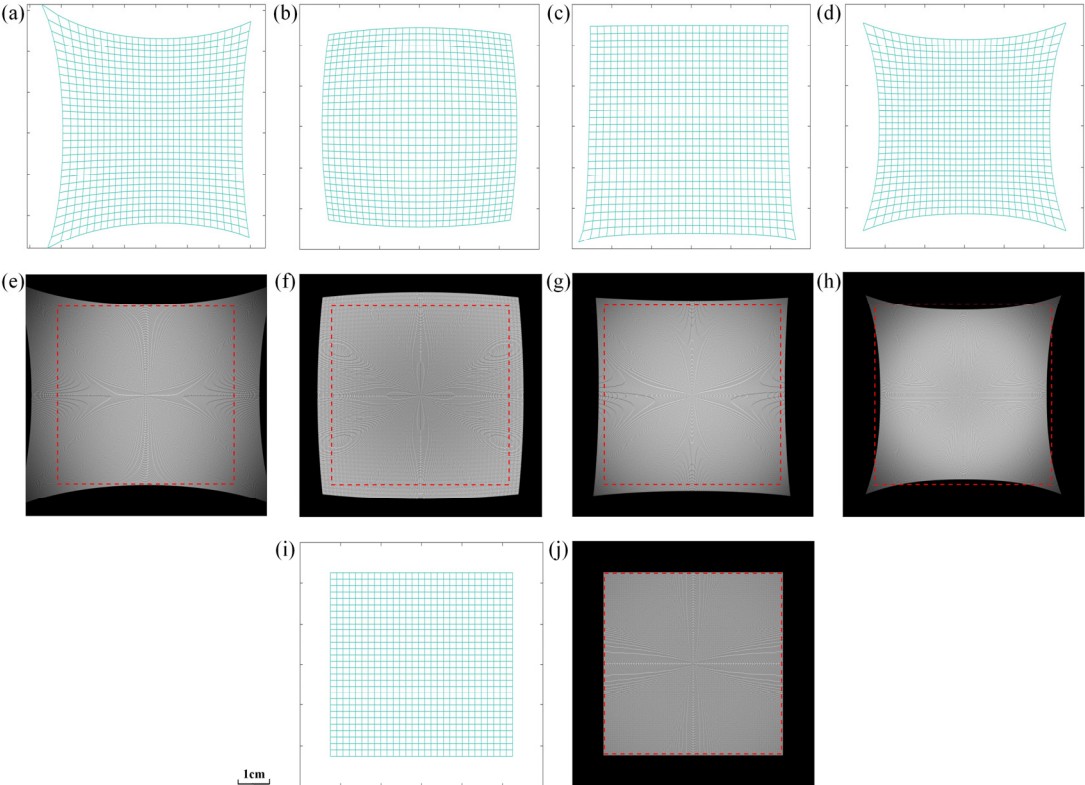

**Figure 6.** Comparison of the effects of the freeform surfaces constructed via the IFPI and other four methods. (**a**) Precise mapping grid on the target surface generated via the V1–V2 method; a 30 × 30 grid is shown. (**b**) Precise mapping grid produced via the linear advection equation method. (**c**) Precise mapping grid produced via the conventional surface construction method. (**d**) Precise mapping grid produced via Feng's method. (**e**) Corresponding irradiance distribution by the V1–V2 method; a 200 × 200 grid is used. (**f**) Corresponding irradiance distribution by the linear advection equation method. (**g**) Corresponding irradiance distribution by the conventional surface construction method. (**h**) Corresponding irradiance distribution by Feng's method. (**i**) Precise mapping grid on the target surface generated via the IFPI method. (**j**) Corresponding irradiance distribution by the IFPI method.

Figure 6e–h,j show the illumination effects of the freeform surface obtained via the five methods above, respectively. Figure 6j shows that the freeform surface obtained via the IFPI method can effectively achieve a uniform illuminating distribution of the desired result in the nonparaxial region. The illuminating distributions produced by the other four methods are nonuniform and distorted as the off-axis angle increases; see the four edges in Figure 6e–h. The results indicate that the IFPI method shows significant improvements compared with the other four methods, particularly in the off-axis region under constant ray mapping.

### 3.3. Complex Image Reproduction via IFPI

The last example involves reproducing an image with intricate details via the IFPI method. The original image is shown in Figure 7a, which is a tree pattern with a square resolution set to $200 \times 200$ pixels. The light source is set as a Gaussian density distribution, and the target is set as the density distribution of Figure 7a. The source-to-target map is solved on the basis of the $L^2$-optimal transport in Equations (2) and (3) via the numerical iteration method in [27], as shown in Figure 7d, which remains unchanged during the iteration. The aperture of the freeform lens is set as a square with a size of $2 \times 2$ mm, the size of the target plane is set as 100 mm $\times$ 100 mm, and the distance between the lens and the target plane is set as 50 mm for nonparaxial situation and 570 mm for paraxial situation, with the off-axis angle of 45° for the nonparaxial situation and 5° for the paraxial situation. The freeform lens is constructed by IFPI with 53 steps converging to a tolerance of $10^{-6}$ in less than 180 s.

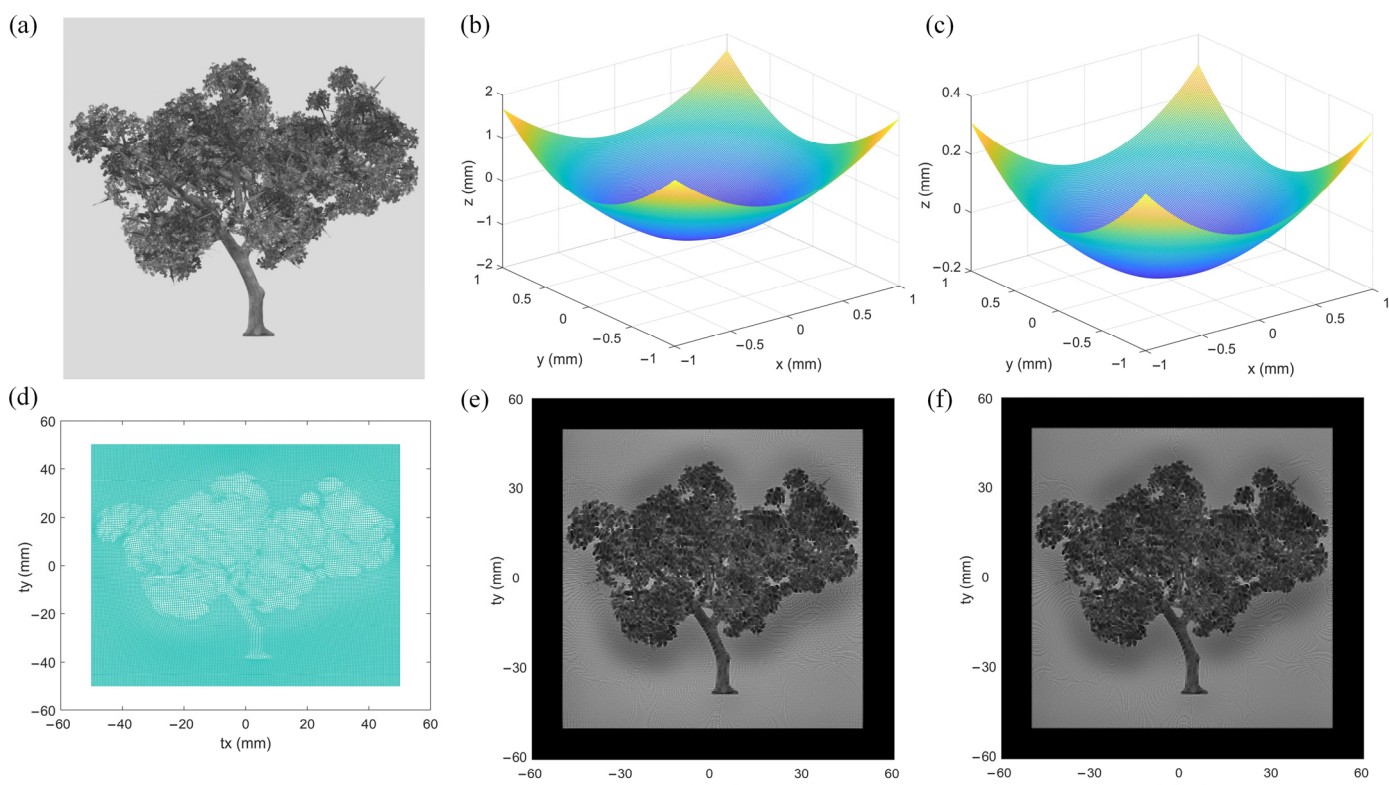

**Figure 7.** The image sample utilizes a tree pattern generated by the FPI method. (**a**) Original image of a tree; (**b**) freeform surface profile built by IFPI for nonparaxial region; (**c**) freeform surface profile built by IFPI for paraxial region; (**d**) preset target map, remained unchanged during the iteration; (**e**) simulation of the illumination distribution in nonparaxial case; (**f**) simulation of the illumination distribution in paraxial case.

Figure 7b,e show the freeform surface profile and the simulated illuminance for the nonparaxial situation, with an RMS of 0.046 mm. On the other hand, Figure 7c,f show the freeform surface profile and simulated illuminance for the paraxial situation, with an RMS of 0.042 mm. It is evident that the achieved results are highly consistent with the preset source-to-target map and illuminance distribution, both in paraxial and nonparaxial regions.

*3.4. Discussion*

Comparison results from Section 3.2 indicate that freeform surface construction methods using paraxial or far-field approximations are usually unable to produce integrable ray mapping outcomes in nonparaxial situations. If these methods are used to construct freeform surfaces, it is essential to implement processes that adjust the ray mapping to meet the integrability condition. The IFPI method continuously iterates the BGE equation to achieve the target mapping without altering the mapping itself. This process involves the surface adapting to the integrable mapping rather than the mapping adapting to the surface.

For the complex image reproduction shown in Figure 7, even though the design methods that enhance ray mapping can be effectively implemented, the complexity of mapping calculations may nonlinearly increase as the grid becomes more complex. In contrast, the IFPI method requires calculating the mapping only once, making the surface iteration process linear. Additionally, as demonstrated in the results of Section 3.1, its convergence speed is both fast and stable. This marks a significant difference between the IFPI method and other integrable design methods.

The current IFPI method proposed in this paper is implemented using Cartesian coordinates (x, y) with a collimated beam. This method can also be extended to other orthogonal coordinate systems, such as polar coordinates (r, $\varphi$), to accommodate the domain of point light sources. The relevant mathematical treatment can be found in [15]. We will address this point in our future work.

## 4. Conclusions

We propose an efficient IFPI method to address the integrability problem in the ray mapping method for freeform lens design. An equivalent form of the integrable condition is substituted into Snell's law to form a balanced expression of the freeform surface of the FPI type. We prove that such FPI equations have a unique solution, which is determined by the geometric properties of the freeform surface and is independent of the source–target map. Benefiting from heuristic processes, the method can calculate a superior freeform surface in the nonparaxial region using a fixed source–target map based on optimal transport. The iterative feature results in a higher convergence speed in the off-axis region. We demonstrated the efficiency and robustness of the proposed method by designing freeform lenses to generate complex illuminated images.

**Author Contributions:** Conceptualization, Y.L.; Software, J.C.; Validation, J.C.; Resources, H.H.; Writing—original draft, J.C.; Writing—review & editing, Y.Z.; Project administration, Y.Z. All authors have read and agreed to the published version of the manuscript.

**Funding:** National Natural Science Foundation of China (62005044); Research Fund of Guangdong-Hong Kong-Macao Joint Laboratory for Intelligent Micro-Nano Optoelectronic Technology (No. 2020B1212030010).

**Institutional Review Board Statement:** Not applicable.

**Informed Consent Statement:** Not applicable.

**Data Availability Statement:** The simulated algorithm and data are all included in the article.

**Conflicts of Interest:** The authors declare no conflict of interest.

## Abbreviations

The following abbreviations are used in this manuscript:

FPI     Fixed-point iteration
IFPI    Implicit fixed-point iteration
BGE    Balanced gradient equation
RMS    Root mean square
PDE    Partial differential equation

## Appendix A. Derivation of BGE

Notes: Subscripts below the symbol $\partial$ mean partial derivative; subscripts below a vector mean components along the coordinate axis.

Expanding Equation (5) in scalar form and using the first and second formula to divide the third one, we can obtain

$$\begin{cases} AN_x = n_o O_x - n_i I_x \\ AN_y = n_o O_y - n_i I_y \\ AN_z = n_o O_z - n_i I_z \end{cases} \Rightarrow \begin{cases} \frac{N_x}{N_z} = \frac{n_o O_x - n_i I_x}{n_o O_z - n_i I_z} \\ \frac{N_y}{N_z} = \frac{n_o O_y - n_i I_y}{n_o O_z - n_i I_z} \end{cases}. \tag{A1}$$

For the collimated beam, the unit vector of incident light is $I = (0, 0, 1)$. Substituting the expression of $N_x, N_y$, and $N_z$ in Equation (4) into Equation (A1), and letting $I_x = 0, I_y = 0, I_z = 1$, and $n_0 = 1$, Equation (A1) can be converted into the following form:

$$\begin{cases} n_i \cdot \partial_x Z = O_x + O_z \cdot \partial_x Z \\ n_i \cdot \partial_y Z = O_y + O_z \cdot \partial_y Z \end{cases}. \tag{A2}$$

A reduction vector $\mathbf{O}'$ is introduced with the following form:

$$(O_x', O_y', O_z') = (O_x, O_y, O_z)/n_i.$$

By dividing both sides of Equation (A2) by n and substituting the reduced vector $\mathbf{O}'$ in it, then Equation (6) can be obtained:

$$\begin{cases} \partial_x Z = O_x' + O_z' \cdot \partial_x Z \\ \partial_y Z = O_y' + O_z' \cdot \partial_y Z \end{cases}.$$

## Appendix B. Derivation of Equation (8)

The distance $\rho$ from points on the freeform surface $\vec{\mathbf{G}} = (x, y, z)$ to the target surface $\vec{\mathbf{T}} = (t_x, t_y, t_z)$ is

$$\rho = \left| \vec{\mathbf{T}} - \vec{\mathbf{G}} \right| = \sqrt{(t_x - x)^2 + (t_y - y)^2 + (t_z - z)^2}.$$

According to the geometric relationship, the unit outgoing vector $\vec{\mathbf{O}}$ has the form

$$\vec{\mathbf{O}} = \frac{\vec{\mathbf{T}} - \vec{\mathbf{G}}}{\rho} \Rightarrow \begin{cases} O_x = \frac{t_x - x}{\rho} \\ O_y = \frac{t_y - y}{\rho} \\ O_z = \frac{t_z - z}{\rho} \end{cases}.$$

From the above relationship, it is easy to find that the gradient of $\rho$ is equal to $-\vec{\mathbf{O}}$:

$$\nabla\rho = (\partial_x\rho, \partial_y\rho, \partial_z\rho) = (-O_x, -O_y, -O_z). \tag{A3}$$

The total differential of $\rho$ is given as follows:

$$d\rho = \partial_x\rho\,dx + \partial_y\rho\,dy + \partial_z\rho\,dz.$$

Integrating both sides of the above equation, we can obtain another expression of $\rho$:

$$\rho = \int \partial_x\rho\,dx + \int \partial_y\rho\,dy + \int \partial_z\rho\,dz = -\left[\int O_x dx + \int O_y dy + \int O_z dz\right] + C_1, \tag{A4}$$

where $C_1$ is a constant.

The total differential of Z is as follows:

$$dZ = \partial_x Z\,dx + \partial_y Z\,dy.$$

Integrating both sides of the above equation,

$$Z = \int \partial_x Z\,dx + \int \partial_y Z\,dy + C_2, \tag{A5}$$

where $C_2$ is a constant.

Substituting Equations (6) and (A4) into the right-hand side of Equation (A5), we have the following:

$$\begin{aligned}
Z &= \int (O_x' + O_z'\partial_x Z)\,dx + \int (O_y' + O_z'\partial_y Z)\,dy + C_2 \\
&= \int O_x'\,dx + \int O_y'\,dy + \left[\int O_z'\partial_x Z\,dx + \int O_z'\partial_y Z\,dy\right] + C_2 \\
&= \int O_x'\,dx + \int O_y'\,dy + \int O_z'\,dz + C_2 \\
&= \left(\int O_x dx + \int O_y dy + \int O_z dz\right)/n_i + C_2 \\
&= -\rho/n_i + C,
\end{aligned}$$

where $C = C_1/n_i + C_2$ is a constant, and $\int O_z'\,dz = \int O_z'\partial_x Z\,dx + \int O_z'\partial_y Z\,dy$.

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
