# Peer review of "Adaptive Construction of Freeform Surface to Integrable Ray Mapping Using Implicit Fixed-Point Iteration"

_photonics, doi:10.3390/photonics12020134_

Round 1

Reviewer 1 Report

Comments and Suggestions for Authors

Comments on the manuscript titled “Adaptive construction of freeform surface to integrable ray mapping using implicit fixed point iteration” Manuscript ID: photonics-3418139.

 This manuscript presents a method for constructing a freeform surface that satisfies the integrability condition and Snell’s law, given a specific invariant source-target map. It uses a fixed-point iteration method for this purpose.

The research topic addressed by the authors in this manuscript, in general terms, is well organized. However, the manuscript requires revisions due to some issues and minor errors. The points that need improvement are listed below:

 1.     On page 3, lines 93 -98, there is a paragraph out of context. It must be removed. It is a comment from some other reviewer.

2. The paragraph described in lines 100-107 is not represented in Figure 1. It should be enhanced to include the description of that paragraph.

3. On page 4, the paragraph from lines 136-138 is confusing and should be rewritten.

 4. On page 4, line 159, the distance functional ρ(x,y,z) Equation is unnumbered.

 5. In Figure 4(a), which are the units in the vertical axis?

 6. Finally, in Figure 6, which shows the results, I do not see what the contribution of the research is.

In my opinion, a much more exhaustive and not so simple and superficial discussion of the results obtained, particularly of Figure 6, should be made so that the reader can appreciate the achievements of the research. The rest of the text of the manuscript describes the technical details that are important to include but do not constitute material of interest to the general reader.

Comments on the Quality of English Language

Overall it is well written but there are a couple of paragraphs that need to be rewritten so that the research can be understood.

Author Response

1. Summary

Thank you very much for taking the time to review this manuscript. We appreciate your suggestions on the manuscript and have made revisions to address the previous shortcomings. Please find the detailed responses below, along with the corrections highlighted in the re-submitted files.

2. Point-by-point response to Comments and Suggestions for Authors

Comments 1: On page 3, lines 93 -98, there is a paragraph out of context. It must be removed. It is a comment from some other reviewer.

Response 1: Thank you for pointing out the mistake in our previous manuscript. We have removed this paragraph in the revised manuscript.

Comments 2: The paragraph described in lines 100-107 is not represented in Figure 1. It should be enhanced to include the description of that paragraph.

Response 2: Thank you for pointing out the omission of Figure 1. In new manuscript, Figure 1 has been enhanced to include all the variables of the paragraph in lines 93-99 (lines 100-107 in last version).

Comments 3: On page 4, the paragraph from lines 136-138 is confusing and should be rewritten.

Response 3: To relieve confusion, the original paragraph from lines 136-138 on page 4 has been rewritten as follows: “Snell's Law in Eq. (5) reveals the relationship between normal vector N, outgoing light vector O, and incident light vector I at every point on a surface. Eq. (6) is derived from Snell's law, so it expresses the relationship between the corresponding physical quantities of surface coordinates Z and outgoing vector O at each point on the surface.“ The changes can be found on page 4, lines 129-132.

3. Comments 4: On page 4, line 159, the distance functional ρ(x,y,z) Equation is unnumbered.

Response 4: Thank you for pointing this out. the distance functional ρ(x,y,z) Equation has been numbered, and the subsequent formula numbers have been rearranged accordingly.

5. Comments 5: In Figure 4(a), which are the units in the vertical axis?

Response 5: The units in the vertical axis of Figure 4(a) are millimeters (mm) and is labeled in Fig. 4(a) in the revised version.

Comments 6:  Finally, in Figure 6, which shows the results, I do not see what the contribution of the research is.

In my opinion, a much more exhaustive and not so simple and superficial discussion of the results obtained, particularly of Figure 6, should be made so that the reader can appreciate the achievements of the research. The rest of the text of the manuscript describes the technical details that are important to include but do not constitute material of interest to the general reader.

Response 6: Thank you for your suggestion. We have added a new section (3.4 Discussion) that provides a detailed analysis of the results, highlighting the effectiveness of the algorithm and emphasizing the contributions of our method. The modification can be found in lines 351 - 370 of the revised manuscript.

Reviewer 2 Report

Comments and Suggestions for Authors

The authors propose the implicit fixed point iteration method to address the integrability in the ray mapping that is utilized in the freeform lens design. Both the geometry definition and the ray mapping are not changed during the iterating progress, promising the efficiency of the proposed design.

However, there are some concerns from the reviewer as follows:

1. As shown in Fig. 5, the RMS changes sharply with the iteration going on, is it possible to give the relatively good choice iterating time for the special situation?

2. To address the efficiency of the proposed design, the extra quantified comparisons among the other ones are supposed in the revised version.

Comments on the Quality of English Language

None

Author Response

1. Summary

Thank you very much for taking the time to review this manuscript. We appreciate your suggestions on the manuscript and have made revisions to address the previous shortcomings. Please find the detailed responses below, along with the corrections highlighted in the re-submitted files.

2. Point-by-point response to Comments and Suggestions for Authors

Comments 1: As shown in Fig. 5, the RMS changes sharply with the iteration going on, is it possible to give the relatively good choice iterating time for the special situation?

Response 1: In Fig. 5, the number of iterations is determined by a predefined minimum RMS threshold. The smaller the convergence threshold, the higher the surface accuracy obtained, but the longer the iterating time take. In our experience, 0.001mm almost meets most surface accuracy requirements.

In special scenarios, we can adjust this threshold to achieve a balance between computational efficiency and solution precision. In the revised manuscript, we added random jitter to the target grid point positions to assess the algorithm's stability and iteration time. Throughout 50 different random grid tests, the convergence trend and iteration time for various target grids were found to be relatively stable, as depicted in Fig. 5(b) and 5(c). The modification can be found in lines 280 - 288 of the revised manuscript.

Comments 2: To address the efficiency of the proposed design, the extra quantified comparisons among the other ones are supposed in the revised version.

Response 2: Thank you for point this out. We agree with this comment. Therefore, we have added a sub-section “3.2 Comparison with other freeform surface construction method” in the revised version. The modification can be found in lines 289 - 324 of the revised manuscript.

Reviewer 3 Report

Comments and Suggestions for Authors

The manuscript titled “Adaptive construction of freeform surface to integrable ray mapping using implicit fixed point iteration” proposes a freeform surface construction method and introduces the specific design methodology, which may be expected to facilitate the design of off-axis optical systems. This study demonstrates the feasibility of this methodology through theoretical derivations as well as the application of two design cases. The manuscript benefits from clear figures and flowcharts, enhancing reader comprehension. The overall logic is relatively clear. However, the manuscript could be improved in several ways that would help readers more fully understand this method.

1. In this manuscript, it is stated that the proposed free-form surface construction method is applicable to collimated beam shaping, so how effective is the proposed method when the beam is divergent? Are there any potential improvements that can be added to the proposed method for diverging beams?

2. In “3.Design examples” Section, could the authors illustrate the optical performance of the system with more diagrams? Additional data and detailed graphical representations of critical optical system evaluation metrics—such as the Modulation Transfer Function (MTF), spot diagrams, and point spread functions (PSF)—would significantly enrich the presentation. These evaluation metrics offer a clear visualization of the improvements in optical performance resulting from the optimization process.

3. The abstract asserts that the robustness and high efficiency of the proposed method are demonstrated with several design examples. However, the manuscript's treatment of the design examples and results lacks the explicit demonstration or substantiation of the proposed method's robustness and high efficiency as claimed in the abstract. Could the authors elaborate on how the proposed method's robustness and efficiency are superior to existing methodologies? A more rigorous comparative analysis, bolstered by quantitative metrics, would be instrumental in enhancing the credibility.

4. What the first paragraph on page 3 of the manuscript, located below Figure 1. is trying to convey? It is suspected that this content may have been inadvertently included and may not be relevant to the subject. If this is indeed the case, please review and amend all similar instances within the manuscript.

5. On line 164 of the manuscript, the expression appears somewhat unclear, and the sentences seem logically inconsistent. The phrase "it is not easy to solve them directly" may be intended to convey the opposite, that is, "it is easy to solve them directly." Could the authors kindly elucidate the intended meaning of the passage? On line 164 of the manuscript, the phrase "The size of the freeform lens is set as 2 mm × 2 mm" raises concerns regarding clarity and specificity, particularly in the context of a three-dimensional object such as a lens. The term "size" is ambiguous and does not convey whether it refers to the lens's diameter, aperture, or another dimensional attribute. Given the three-dimensional nature of lenses, it is crucial to specify the intended dimension for accurate interpretation. If "size" refers to the lens's aperture, it should be explicitly stated.

6. The manuscript contains grammar errors and issues with standardization, which necessitate careful review and improvement. For instance, it is advised that all vectors be presented in a vectorial form to facilitate comprehension for the readers; according to academic standards, the “law” in “Snell's law” should be capitalized and the appropriate formulation is 'Snell's Law'; on line 214, "as follow" should be corrected to "as follows", on line 164 of the manuscript "Because the solution of Eq. (9) has one common set of Zv." appears to be incomplete and lacks clarity, and so on. It is noteworthy that the grammar errors in the manuscript are not limited to those mentioned above. The authors are suggested to conduct an in-depth review and make the necessary corrections.

Author Response

1. Summary

Thank you very much for taking the time to review this manuscript. We appreciate your suggestions on the manuscript and have made revisions to address the previous shortcomings. Please find the detailed responses below, along with the corrections highlighted in the re-submitted files.

2. Point-by-point response to Comments and Suggestions for Authors

Comments 1: In this manuscript, it is stated that the proposed free-form surface construction method is applicable to collimated beam shaping, so how effective is the proposed method when the beam is divergent? Are there any potential improvements that can be added to the proposed method for diverging beams?

Response 1: Your question is very insightful and revelatory. We believe that introducing polar coordinates will enable our method to extend to cases involving divergent beams or point light sources. We will explore this situation in our future work. We have included a paragraph in the revised manuscript (lines 366-370) to address this issue.

Comments 2: In “3.Design examples” Section, could the authors illustrate the optical performance of the system with more diagrams? Additional data and detailed graphical representations of critical optical system evaluation metrics—such as the Modulation Transfer Function (MTF), spot diagrams, and point spread functions (PSF)—would significantly enrich the presentation. These evaluation metrics offer a clear visualization of the improvements in optical performance resulting from the optimization process.

Response 2: Your opinion is very professional and useful. MTF, spot diagrams, and PSF can effectively evaluate the point-to-point imaging quality in imaging optics. However, the methods proposed in this manuscript are mainly applied to beam shaping and illumination in non-imaging optics. In non-imaging optics, the light emitted from the light source does not necessarily need to converge into a single point on the target surface. Therefore, it seems that these functions cannot be easily applied here to evaluate non-imaging optical design results. We referred to reference [25] and used Root-mean-square (RMS) value as a measure of the quality of our design results, and added more diagrams of RMS to enrich the presentation, as shown in Fig. 5 in the revised manuscript.

Comments 3: The abstract asserts that the robustness and high efficiency of the proposed method are demonstrated with several design examples. However, the manuscript's treatment of the design examples and results lacks the explicit demonstration or substantiation of the proposed method's robustness and high efficiency as claimed in the abstract. Could the authors elaborate on how the proposed method's robustness and efficiency are superior to existing methodologies? A more rigorous comparative analysis, bolstered by quantitative metrics, would be instrumental in enhancing the credibility.

Response 3: Thank you for your suggestion. We have added sub-section 3.2 (pages 9, lines 289 - 324) to the revised manuscript to elaborate on the efficiency of the proposed method in comparison to existing methodologies. The robustness of our method has been tested by introducing 50 sets of random grids, and the results have been included in Fig. 5 of the revised manuscript (lines 280 – 288).

Comments 4: What the first paragraph on page 3 of the manuscript, located below Figure 1. is trying to convey? It is suspected that this content may have been inadvertently included and may not be relevant to the subject. If this is indeed the case, please review and amend all similar instances within the manuscript.

Response 4: Thank you for pointing out the mistake in our previous manuscript. We have removed this paragraph in the revised manuscript. We have reviewed the new manuscript to ensure that there is no similar case.

Comments 5: On line 164 of the manuscript, the expression appears somewhat unclear, and the sentences seem logically inconsistent. The phrase "it is not easy to solve them directly" may be intended to convey the opposite, that is, "it is easy to solve them directly." Could the authors kindly elucidate the intended meaning of the passage? On line 164 of the manuscript, the phrase "The size of the freeform lens is set as 2 mm × 2 mm" raises concerns regarding clarity and specificity, particularly in the context of a three-dimensional object such as a lens. The term "size" is ambiguous and does not convey whether it refers to the lens's diameter, aperture, or another dimensional attribute. Given the three-dimensional nature of lenses, it is crucial to specify the intended dimension for accurate interpretation. If "size" refers to the lens's aperture, it should be explicitly stated.

Response 5: Thank you for pointing this out. We agree with your rigorous opinions on the grammar and definitions of these expressions. We have corrected the expression on line 164 of “Although  and  are both unknown variables in Eq. (7), it is not easy to solve them directly.” to “Because  and  are both unknown variables in Eq. (7), it is difficult to solve them using just a single equation.”. The changes can be found on page 5, lines 159-160.

The expression of original “The size of the freeform lens is set as 2 mm×2 mm” has been changed into “The aperture of the freeform lens is set as a square with size of 2 mm × 2mm”. The changes can be found on page 10, lines 332.

Comments 6: The manuscript contains grammar errors and issues with standardization, which necessitate careful review and improvement. For instance, it is advised that all vectors be presented in a vectorial form to facilitate comprehension for the readers; according to academic standards, the “law” in “Snell's law” should be capitalized and the appropriate formulation is 'Snell's Law'; on line 214, "as follow" should be corrected to "as follows", on line 164 of the manuscript "Because the solution of Eq. (9) has one common set of Zv." appears to be incomplete and lacks clarity, and so on. It is noteworthy that the grammar errors in the manuscript are not limited to those mentioned above. The authors are suggested to conduct an in-depth review and make the necessary corrections.

Response 6: Thank you for pointing these out.

(1)    We have replaced the vectors in the revised manuscript with vectorial form;

(2)    “Snell's law” has been changed into 'Snell's Law' in the whole manuscript;

(3)    "as follow" has been corrected to "as follows" in the whole manuscript;

(4)    The sentence of "Because the solution of Eq. (9) has one common set of Zv." is changed as follows: “By combining the two coefficient matrices of BGEs into a single matrix H, Eq. (9) utilizes a common set of Zv, thereby resolving the issue of having two different data points as presented in Eq. (6).” on page 6, lines 212-214;

(5)    We conducted a thorough review of the entire manuscript and made necessary corrections, which are highlighted in red.

Round 2

Reviewer 1 Report

Comments and Suggestions for Authors

The authors have addressed all recommendations and corrected errors and missing information in the manuscript. For this reason, I recommend its publication in the current version.

Reviewer 2 Report

Comments and Suggestions for Authors

All the concerns from the reviewer have been addressed and modified in the revised version.